# Novel Tribometer for Coated Self-Lubricating Spherical Plain Bearings in a Vacuum

Zhen Li [1,2], Zhinan Zhang [1,2,*], Qingsong Yong [3,*], Guozheng Ma [4], Aobo Wei [5] and Haidou Wang [4,5]

1  State Key Laboratory of Mechanical System and Vibration, Shanghai Jiao Tong University, Shanghai 200240, China
2  School of Mechanical Engineering, Shanghai Jiao Tong University, Shanghai 200240, China
3  Facility Design and Instrumentation Institute of China Aerodynamics Research and Development Center, Mianyang 621000, China
4  National Key Laboratory for Remanufacturing, Army Academy of Armored Forces, Beijing 100072, China
5  National Engineering Research Center for Remanufacturing, Army Academy of Armored Forces, Beijing 100072, China
*  Correspondence: zhinanz@sjtu.edu.cn (Z.Z.); qingsong_yong@163.com (Q.Y.)

**Abstract:** Coated self-lubricating spherical plain bearings (SSPBs) are a fairly key component of the space-swing mechanism. To examine the operation status and tribological properties of coated SSPBs, a tribometer with a temperature control module in a vacuum condition was developed. The tribometer was mainly composed of a fixture system, reciprocating rotational motion system, environment control system, etc. First, the tribometer was verified with the self-made hydrogenated diamond-like carbon (H-DLC) SSPBs. The sensor signals indicated that the tribometer conformed to the design specifications. Then, the influence of friction heat on the tribological properties of H-DLC SSPBs was analyzed. The results showed that friction torque and temperature increased with the overall test time. Although the temperature had reached 48 °C, the frictional heat had little effect on the H-DLC SSPBs' lifespans. The damage mechanism of H-DLC SSPBs was dominated by abrasive wear and fatigue wear in vacuum conditions.

**Keywords:** tribometer; friction heat; H-DLC film; self-lubrication spherical plain bearings

## 1. Introduction

As a spherical surface contact pair, spherical plain bearings are mainly composed of an inner ring with an outer spherical surface and an outer ring with an inner spherical surface. It has the characteristics of bearing capacity, impact resistance, wear resistance, being maintenance-free, etc., and is widely used in hydraulic, aerospace, aviation, and other oscillating motion fields [1–3]. According to the different bearing lubricating methods, spherical plain bearings can be divided into non-lubricated, grease-lubricated, oil-lubricated, and self-lubricating spherical plain bearings (abbreviated as SSPBs). Compared with the traditional lubricating methods, the SSPBs is the preferred type for vacuum oscillation parts as they do not require sealing devices and additional lubricants. The important feature of SSPBs refers to materials with self-lubricating properties that are coated/adhered on the inner surface of the outer ring, the outer surface of the inner ring, or both. According to the different preparation methods, SSPBs can be classified into two types: fabric SSPBs and coated SSPBs [4,5]. The coating/film preparation process of the coated SSPBs is simple, and its product quality consistency is good, which can be used in oscillation parts that require a low coefficient of friction and high reliability.

For the sake of acquiring good tribological properties in coated SSPBs, relevant scholars have carried out research work on the manufacturing techniques of bearing and self-lubricating materials. Xue et al. [6] presented a new mating structure of SSPBs and optimized the flatness of the contact surface, resulting in an improvement in its contact and

tribological properties. Moreover, Guo et al. [7] found that when the surface roughness of SSPBs substrate is low, the adhesive wear between friction pairs will be strengthened; when the roughness is high, it will lead to greater abrasive wear.

On the other hand, the inherent low-friction properties of the coating are one of the essential factors affecting the self-lubricating performance of SSPBs. The hydrogenated diamond-like carbon(H-DLC) film has excellent tribological properties owing to its special physical structure, which consists of $sp^3$-bonded carbon atoms with diamond characteristics and $sp^2$-bonded carbon atoms with graphite characteristics [8–10]. Li et al. [11] prepared H-DLC, Cr-H-DLC, and B-H-DLC films on an AISI 52100 steel substrate and analyzed the friction and wear properties under a vacuum ball disc reciprocating testing environment. The results showed that all films have a low coefficient of friction and wear rate. Liu et al. [12] investigated the effect of high speed on the superlubricity properties of H-DLC films; they found that the presence of the transfer film plays an important role in maintaining the superlubricity properties of the friction pair. Solis et al. [13] prepared Cr-WC-a-C:H films using the plasma enhanced chemical vapor deposition(PECVD) technique and studied the films' tribological properties under the pin-plate test condition. Owing to the formation of a protective transferred layer, the wear rate decreased as the load increased. Jiadong et al. [14] improved the tribological properties of DLC films by adding a graphite film, which was prepared using a spraying technique.

In summary, by rationally selecting the form and position tolerance of the bearing substrate and DLC films with excellent tribological properties, the properties and service lives of the coated SSPBs can be improved. However, how to evaluate the service performance of coated SSPBs requires further investigation. At present, the common way is to use a bearing testing machine for analysis. Liu et al. [15,16] studied the wear failure mechanism of spherical plain bearings with Mn-P coating under light load and low-frequency swing conditions by oscillating a wear life testing machine for the atmospheric environment. The result showed that the friction torque and temperature signal can be used to judge the bearing performance. Qiu et al. [5] utilized a radial spherical plain bearing testing machine to conduct friction and wear tests on a spherical plain bearing using $MoS_2$/graphite composite coating as the self-lubricating material and found that the SSPBs had the best tribological properties and longest service lives. Furthermore, Xue et al. [17] utilized the heavy-load and low-frequency test machine to analyze the wear mechanism of fabric SSPBs. The test result showed that the wear zone is greatest in the middle of the self-lubricating materials. Beyond that, Lu et al. [18] invented a high-frequency and light-load tribometer in an atmospheric environment and evaluated the service performance of the SSPBs. Liu et al. [19] developed a high-temperature and heavy-load testing machine for SSPBs and obtained the tribological properties of SSPBs at 20 °C and 75 °C.

The above studies show that relevant scholars have developed different types of SSPB testing machines for different bearing application conditions. However, these testing machines are mostly aimed at atmospheric environments, and there is a lack of testing machines for SSPBs in a vacuum environment. Moreover, the literature shows that temperature can affect the tribological properties of self-lubricating materials [19,20]. Heat will inevitably be generated due to friction in the process of SSPB movement. Especially for the diamond-like carbon films with their lower thermal conductivity, their heat will accumulate in the friction zone; this leads to their temperature increasing and causes their tribological properties to decrease [21,22]. Furthermore, this phenomenon will be more prominent in a vacuum. Therefore, a vacuum SSPB tribometer with a temperature control was developed in the present study, which mainly consisted of a reciprocating rotational motion system, bearing fixture system, environment control system, and data acquisition system. In addition, the SSPBs, which were coated with H-DLC as the lubricating material, were verified in a vacuum environment. Moreover, the effect of friction heat on the tribological properties of the coated SSPBs was also analyzed.

## 2. Design and Verification of the Tribometer

### 2.1. Requirement Definition and Overall Design

According to the sample shape, the friction and wear test can be divided into material and part tests. Generally, reciprocating and rotating [23] modes are the primary forms of material testing, which are mainly designed to test the bulk material of the plate shape. Alternatively, using tribometers as parts must meet different size specifications of the moving part, such as a high-temperature tribometer for bushing [24] and life-testing tribometer for bearings [25] and gears [26]. Therefore, to develop a coated SSPBs tribometer in a vacuum environment, it is necessary to clarify its object characteristics, main motion forms, and failure evaluation methods.

The structures of the SSPBs are illustrated in Figure 1. The primary motion form of the SSPBs is a swing motion around the Y-axis. Furthermore, because the sliding surface is spherical, it can also make oblique movements around the X-axis and Z-axis in a certain angle range. Combined with the current research on the motion form of SSPBs [1,18], the tribometer should be able to realize the swing motion for SSPBs, and its loading direction should be along the radial direction. At present, the tribometer used for fabric SSPBs is well-developed. Additionally, its life evaluation standard has been established all over the world [27,28]. Specifically, because of the large thickness of the fabric ($\geq$400 µm), the wear depth of the fabric can be measured by the displacement sensor, and thus the lifespans of the SSPBs can be evaluated. Nonetheless, the thickness of the carbon-based film ($\leq$30 µm) is much lower than that of fabric. Therefore, it is difficult to measure the wear depth of the films by the same method. Through the accumulation of a large number of experimental data, Liu et al. [15,16] found that the variation trend of friction torque and temperature of coated spherical plain bearings has three stages: namely, a running-in phase, long steady-state phase, and severe wear phase. At the end of the coated spherical plain bearing's life, the changes in friction torque and temperature rise almost simultaneously, and it can be inferred that the bearing has failed. Therefore, the lifespans of coated SSPBs can be confirmed by the friction torque and temperature signal. To sum up, the design specifications of the coated SSPBs tribometer are listed in Table 1. To improve the efficiency of vacuum extraction, the vacuum chamber size is selected to be small, which necessitates the design of a compact tribometer. Moreover, the tribometer should be equipped with a temperature control function to explore the effect of friction heat on the tribological properties of the coated SSPBs in the vacuum environment. Therefore, the temperature control is only required to be within the range of 15–30 °C.

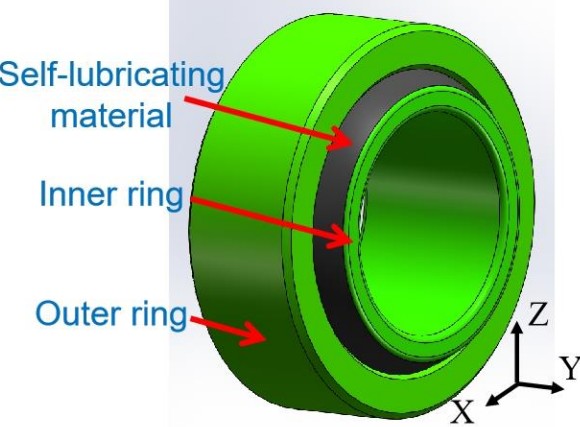

**Figure 1.** The schematic diagram of the SSPBs.

**Table 1.** Design specifications of the coated SSPBs tribometer.

| Parameter | Value |
|---|---|
| Overall dimensions | $\leq 360 \times 250 \times 250$ mm |
| Size range of test bearings | 8~30 mm |
| Load | 0~250 N |
| Friction torque | 0~1 Nm |
| Reciprocating rotation frequency | 0~3 Hz |
| Maximum pendulum angle | $\pm 45°$ |
| Vacuum degree | $\leq 1 \times 10^{-3}$ Pa |
| Temperature control | 15~30 °C |

The structure of the coated SSPBs tribometer is shown in Figure 2. The coated SSPBs tribometer consists of: a fixture system for coated SSPBs that realizes the fixing and positioning of the inner and outer rings; a radial force system that realizes the radial force required for the test through the spring; a reciprocating rotational motion system, which provides the desired motion form; an environmental control system, which meets the requirements of vacuum and temperature control; and a data acquisition system, which realizes the acquisition of temperature, friction torque, and vibration signals during the testing of the coated SSPBs.

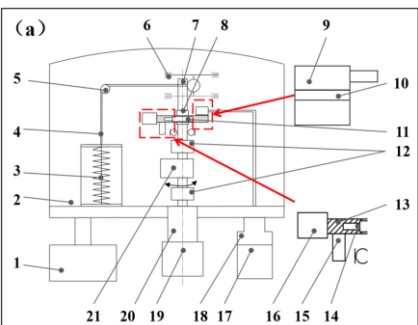 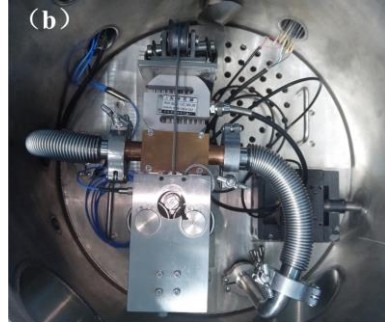

**Figure 2.** The structure of the coated SSPBs tribometer: (**a**) the schematic diagram and (**b**) tribometer from the top view. 1: Vacuum pump; 2: vacuum chamber; 3: spring; 4: wire rope; 5: fixed pulley; 6: sliding table; 7: test shaft; 8: upper fixture; 9: temperature control block; 10: indium film; 11: coated SSPBs; 12: coupler; 13: outer ring fixture; 14: temperature sensor; 15: vibrating sensor; 16: force sensor; 17: temperature control equipment; 18: flange; 19: servo motor; 20: magnetic fluid shaft; 21: friction torque sensor.

*2.2. Bearing Fixture System*

In practical use, the fits between the inner ring and the shaft and between the outer ring and the support pedestal have clearance, and the clearance values are 13–35 μm and 0–25 μm, respectively. Because the friction torque value is used to judge the failure of the bearing, if this method is adopted, it is easy to cause slippage between the inner ring and the shaft and between the outer ring and the support pedestal; this is due to the large friction torque during the test process, which results in an inaccurate measurement. The friction torque is generated due to the frictional resistance caused by the relative movement of the inner ring and the outer ring. To measure the properties of coated SSPBs by their friction torque values, it is necessary to ensure that the inner and outer rings are always in relative motion during the test. Therefore, the inner and outer rings should be separately fixed. In addition, friction heat is generated. The relative motion of the inner and outer rings is located in the contact area, which is difficult to measure. Hence, to measure the friction heat, it is necessary to consider the installation position of the temperature sensor on the outer ring fixture.

Based on the above consideration, the fixture system for the coated SSPBs is shown in Figure 3. The design scheme of the shaft and the upper fixture solves the problem of

fixing the inner ring and test shaft of the coated SSPBs. Considering that the fixed method is realized by friction, to avoid the slippage of the inner ring and test shaft caused by reciprocating motion, the knurling feature was designed in the contact area using the end faces of the inner ring to increase its surface roughness. In addition, a locknut was designed to prevent slipping on the upper fixture. Considering the need for outer ring fixing, the outer ring fixture was equipped with a lock screw, which deforms the fixture by rotating the screw. However, the tightening force depends on the screw-in depth. A torque wrench was used for tightening purposes to keep the force consistent. Normally, the reciprocating motion has a 0° center point, which is the point where the friction heat value is maximum. Moreover, owing to the structure features, it is difficult to monitor the friction heat of the SSPB contact surfaces. Hence, the temperature sensor was placed in the center and near the outer ring (As shown in Figure 3). Additionally, the contact measurement sensor was adopted. Considering the frequent disassembly, the sensor was fixed by an outer ring fixture by the thread.

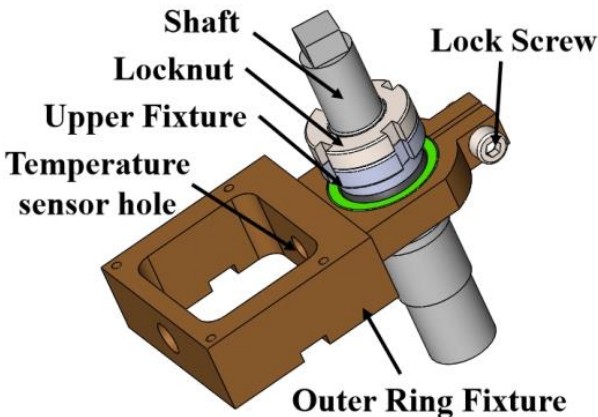

**Figure 3.** The fixture systems of the coated SSPBs tribometer.

*2.3. Radial Force System*

Generally, weight loading [29], hydraulic loading, and spring loading [30] are the main ways to apply the load for the testing machine. The weight-loading method has a simple structure and is only suitable for small load conditions. The hydraulic loading method has the advantage of a large load, but the structure is complex, especially in vacuum environments, which are prone to leakage and contamination of the chamber. In addition to its simple structure, the loading force interval of spring loading was located between the weight loading and hydraulic loading. The spring method was adopted due to the smaller chamber size. Meanwhile, a fixed pulley was used to change the direction of the loading force to adapt to the chamber space. The radial force loading strategy is illustrated in Figure 4. A force sensor was installed at the rear of the outer ring fixture to obtain the force value loaded on the tested bearing. In addition, a two-dimensional sliding table was adopted to reduce the problem of force value fluctuation caused by the misalignment of the axle center.

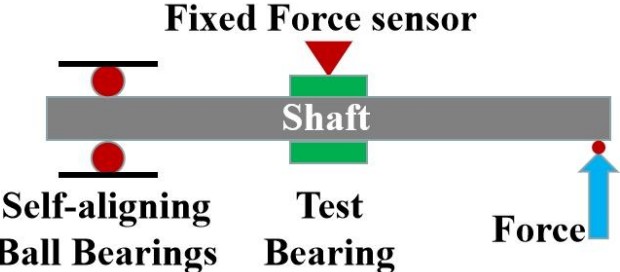

**Figure 4.** The schematic diagram of the radial force loading strategy.

### 2.4. Reciprocating Rotational Motion System

The reciprocating rotational motion can be realized by the four-bar linkage mechanism [31,32], oscillating hydraulic cylinder method [33], and direct drive of the motor. The advantage of the four-bar linkage method is that the structure is simple, and the performance requirements of the motor are low. However, it is difficult to conduct a continuous adjustment of the swing motion. As a type of hydraulic application, the oscillating hydraulic cylinder has the characteristics of large driving torque and large equipment size. Nevertheless, during long-term use, oil leakage will inevitably occur. The direct drive of the motor has a simple structure, which can realize the continuous adjustment of frequency and angle. Additionally, its applications are mostly in low frequency and light load conditions. Owing to the low frequency and light load application situation of the coated SSPBs, the last-mentioned scheme was adopted. By selecting the magnetic fluid seal shaft, the motor could be moved to the outside of the vacuum chamber, which not only reduces the space occupied inside the vacuum chamber but also selects the servo motor as the power source.

### 2.5. Environment Control System

Coated SPBBs are suitable for use in a vacuum environment. To more accurately simulate the actual working conditions, the current mature vacuum environment control equipment was used. The primary-only pump selects a dual-stage DC brushless rotary vane vacuum pump, and its maximum pumping rate can reach 30 m$^3$/h (Value Corp, Taizhou, Zhejiang, China). The secondary pump is an oil-lubricated molecular pump with a pumping rate of up to 1200 L/s under a N$_2$ environment and an ultimate pressure of up to $6 \times 10^{-6}$ Pa (KYKY Corp, Haidian, Beijing, China). The basic vacuum environment preparation process is as follows: First, the chamber is initially vacuumized with a mechanical pump until the air pressure falls below 10 Pa. Then, the molecular pump is turned on to further pump the low-pressure environment. Finally, the chamber reaches a high-vacuum environment [34].

To neutralize the temperature rise caused by friction heat, it is necessary to export and dissipate the heat. Moreover, the heat can only be transferred through heat conduction and heat radiation in a vacuum environment. Based on the above reasons, the design scheme of the temperature control block was used in the tribometer. By controlling the temperature of the outer ring fixture, the temperature of the SSPBs is relatively stable during the testing process. By cooling the liquid, the friction heat is removed from the temperature control block that acts as a heat transfer. More specifically, the liquid temperature control equipment is installed outside the vacuum chamber, and liquid flows into the block through flanges and pipes, thereby realizing the temperature control of the outer ring fixture. Additionally, an indium film 0.2 mm thick was added between the contact surface of the block and the outer ring fixture to improve the heat conduction efficiency and reduce the interface thermal resistance.

### 2.6. Data Acquisition System

During the operation of the testing machine, the operation status of the equipment and the tested bearing should be monitored by different sensors in real time, such as the temperature sensor, torque sensor, force sensor, and vibration sensor. The measurement ranges and accuracies of the different sensors are shown in Table 2. To ensure that the operating signal is within the normal range and to protect the sensor (especially the torque sensor), it is necessary to set a threshold value. When the signal value reaches the threshold value, the tribometer must be stopped. Moreover, each sensor needs to be calibrated to ensure the relative accuracy of the measured data. Because this tribometer adopts the motor direct drive method, the parameters such as the swing angle, reciprocating frequency, and running time must be set through a control program.

**Table 2.** The measurement ranges and accuracies of the different sensors.

| Sensor | Range | Accuracy |
| --- | --- | --- |
| Temperature sensor | −50~200 °C | ±(0.15 + 0.002 | Temperature | ) |
| Force sensor | 0~500 N | 0.03%FS |
| Torque sensor | 0~±1 Nm | ±0.2%FS |
| Vibration sensor | 0~±50 g | 0.002 g |

Figure 5 shows the data acquisition system and control strategy of the coated SSPBs tribometer. The hardware, based on the above consideration, consisting of a 16-bit A/D acquisition card (Yanhua Corp, Suzhou, Jiangsu, China), signal regulation board (Keyuan Corp, Changping, Beijing, China) and servo motor driver were used to collect sensor signals in real time and to control the motor, respectively. In the software design and architecture, the operating software was developed based on the Lab Windows/CVI 2012 software produced by NI Company, which can display the collected sensor signals in real time in the form of images and can set thresholds value and operating parameters.

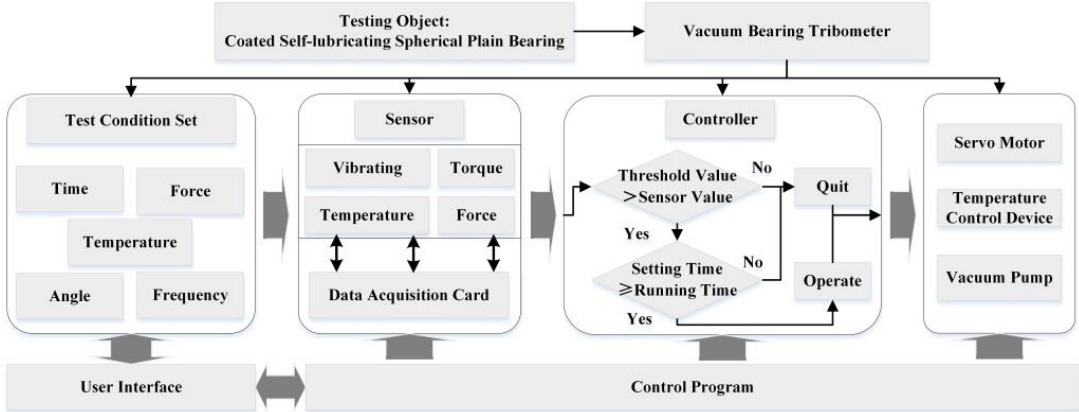

**Figure 5.** The data acquisition system and control strategy of the coated SSPBs tribometer.

*2.7. Tribometer Test*

The hydrogenated diamond-like carbon films show a higher surface quality and excellent tribological properties that are prepared by the unbalanced magnetron sputtering system, which can be used as a lubricating layer of spherical plain bearings without post-processing. Therefore, the H-DLC self-lubricating coating was prepared on the relative sliding surface of the φ17 spherical plain bearing. The outer and inner rings were made of AISI 52100 steel (As shown in Figure 6). In addition, the structure and preparation process of H-DLC films are shown in the literature [11]. The radial and axial clearances of the H-DLC SSPBs are 44.4 ± 0.67 μm and 160 ± 5 μm, respectively.

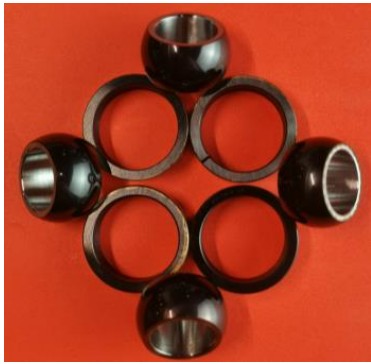

**Figure 6.** The inner and outer rings coated with H-DLC film.

Preliminary validation of the H-DLC SSPBs tribometer was conducted. The performance test duration was 30 min at an initial temperature of 25 °C, reciprocating frequency of 0.5 Hz, load of 175 N, and vacuum pressure of 5 Pa. The motion schematic and contact area on the self-lubricating coating are illustrated in Figure 7. The wear marks on the contact area of the self-lubricating coating are not obvious. However, there are slight scratches, which may be caused by microscopic protrusions on the contact surface. Moreover, the wear morphology is symmetric from the center of motion. The sensor data collected during the experiment is shown in Figure 8. The absolute value of the vibration signal gradually increased with the increase in time; the temperature signal also showed the same variation trend. Owing to the short duration time, the change in amplitude of the above signal was small. The load signal smoothly changed, showing a sinusoidal variation trend with the movement. The average loading force was 174 N, and the error was less than 1%. In addition, the torque signal also exhibited a sinusoidal signal that varied with the form of the motion. The above-mentioned verification test showed that the tribometer meets the requirements of the previous design indicators.

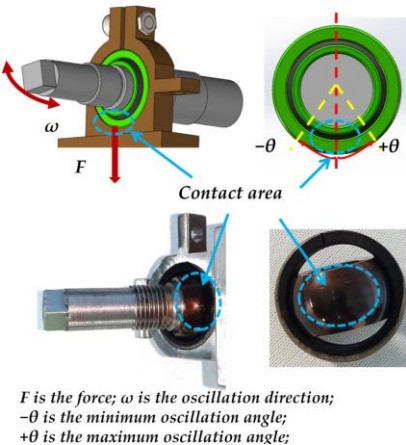

*F is the force; ω is the oscillation direction;*
*−θ is the minimum oscillation angle;*
*+θ is the maximum oscillation angle;*

**Figure 7.** The motion schematic and contact area of the coated SSPBs.

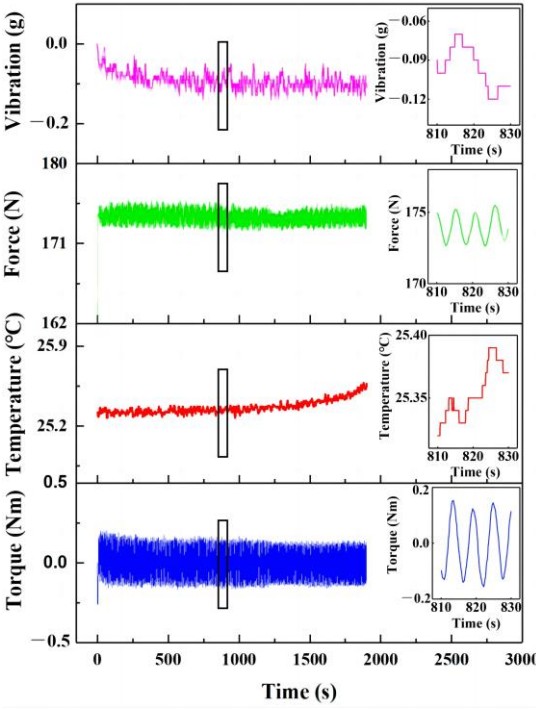

**Figure 8.** The measured sensor data during the experiment.

## 3. Result and Discussion

### 3.1. Coated SSPBs Preparation

The H-DLC SSPBs with hydrogenated diamond-like carbon materials as the self-lubricating coating were used as the test object. To remove the oil and other pollutants before the vacuum test, the H-DLC SSPBs were cleaned in an alcohol solution for 30 min. In this paper, the influence of friction heat on the H-DLC SSPBs' tribological performance was studied. Considering room temperature and vacuum equipment adaptability, the H-DLC SSPBs of the experimental group (hereafter referred to as sample B) were treated at a constant temperature of 25 °C in a vacuum environment. The normal vacuum test was the control group (hereafter referred to as sample A). In addition, the generation of friction heat is related to the speed, load, and coefficient of friction [35]. Therefore, to more clearly analyze the effect of friction heat on the tribological properties of the H-DLC SSPBs, the test time was set to 20 h. The experiment parameters of the coated SSPBs are illustrated in Table 3.

**Table 3.** The experiment parameters of the coated SSPBs.

| Experiment Parameter | Control Group (Sample A) | Experimental Group (Sample B) |
|---|---|---|
| Load | 185 N | |
| Reciprocating frequency | 2 Hz | |
| Reciprocating Angle | 15° | |
| Test Time | 20 h | |
| Initial/steady Vacuum Degree | 5 Pa/0.2 Pa | |
| Acquisition Frequency | 2 Hz | |
| Control Friction Heat | No | Yes (25 °C) |

### 3.2. Tribological Properties

Figure 9 shows the sensor signal of temperature and friction torque of sample A and sample B with or without temperature control conditions. It can be seen from Figure 9a that the temperature first slowly increased, quickly increased, had a relative stability, and finally increased. After 20 h of vacuum testing, the temperature of the H-DLC SSPBs reached approximately 48 °C. This phenomenon indicates that the SSPBs generated heat during the vacuum test. Compared with Figure 9a, the temperature control maintained a relative value (Figure 9c). We must recognize the fact that this temperature value fluctuates, which is caused by the intermittent operation of the temperature control device. Generally speaking, friction heat is related to the pressure, speed, and coefficient of friction. Therefore, under the condition of constant pressure and speed, the amount of friction heat generated is positively correlated with the coefficient of friction. It is shown that the lubricating performance of the H-DLC SSPBs decreased during the severe rising stage. As shown in Figure 9b,d, the friction torque of the H-DLC SSPBs first stabilized, then increased, and finally experienced drastic changes with the increase in time. The above trend proves that the lubrication performance of H-DLC SSPBs is good in the early stage. Nonetheless, as the number of reciprocating swings increases, the lubricating materials are gradually consumed, resulting in a gradual deterioration of the lubrication and an eventual bearing failure. Based on the above analysis, the average lifespans of H-DLC SSPBs under different working conditions is about 80,000 and 85,000 cycles, respectively. The literature [36,37] shows that H-DLC films have better anti-wear properties in an environment below 100 °C. Therefore, it can be concluded that frictional heat has little effect on H-DLC SSPBs' lifespans.

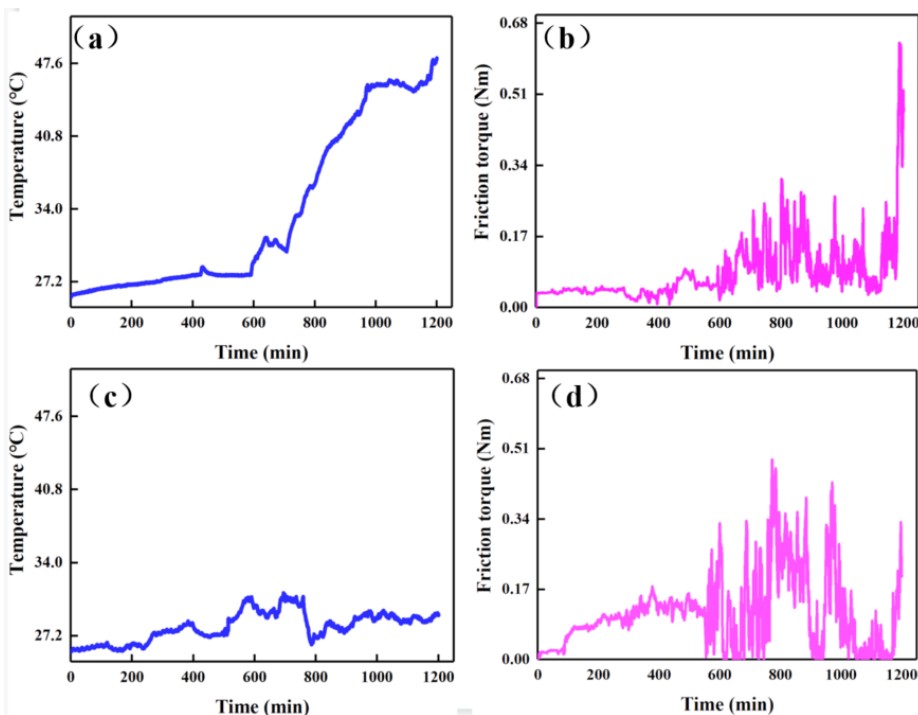

**Figure 9.** The sensor signals: (**a**) temperature of sample A; (**b**) friction torque of sample A; (**c**) temperature of sample B; (**d**) friction torque of sample B.

The optical graphs and EDS analysis of the wear scar of sample A and sample B are shown in Figure 10. As shown in Figure 10a,g, the H-DLC self-lubricating films in the relative sliding area are removed. Therefore, the bright white steel color is exposed, indicating that the H-DLC SSPBs can judge failure under the above working test conditions. There are a large number of groove morphologies in both bearings, which can be found along the sliding direction (Figure 10b,h). During the movement, there are tiny peaks and small particles that behave similar to a dull knife sliding on the contact surface. Consequently, the material is squeezed, stacked, and finally formed into a plowing morphology. Based on the above analysis, the failure forms are caused by abrasive wear. As seen in Figure 10c, a partially flat area (red circle) appeared. It can be inferred that the abrasive particles are squeezed into the substrate and are plastically deformed during the reciprocating motion. As shown in Figure 10d,i, the wear zone of sample A and sample B exhibits the morphological feature of the crack. Owing to the presence of hard particle phases inside the AISI 52100 steel, large shear stresses are generated around these hard phases under frequent reciprocating movements. The crack gradually expands and finally causes the material to exfoliate with the increase in time. Furthermore, the exfoliation would further participate in the wear process and accelerate the wear process of the material. The EDS analyses of the wear topographies of sample A (Figure 10e) and sample B (Figure 10k) are shown in Figure 10f,l. The flat area (Figure 10c) was mainly composed of Fe elements, which proves that the characteristic is from the wear debris of the bearing substrate materials. The reason for its generation is that the debris is flattened and adhered to the relative motion area under the action of force. Moreover, there are long strips of wear scar in both sample A (Figure 10e) and sample B (Figure 10k), which mainly consisted of Fe and C elements. The increase in C content may be due to the transfer of carbon from the H-DLC film to the surface of the substrate material during the friction process.

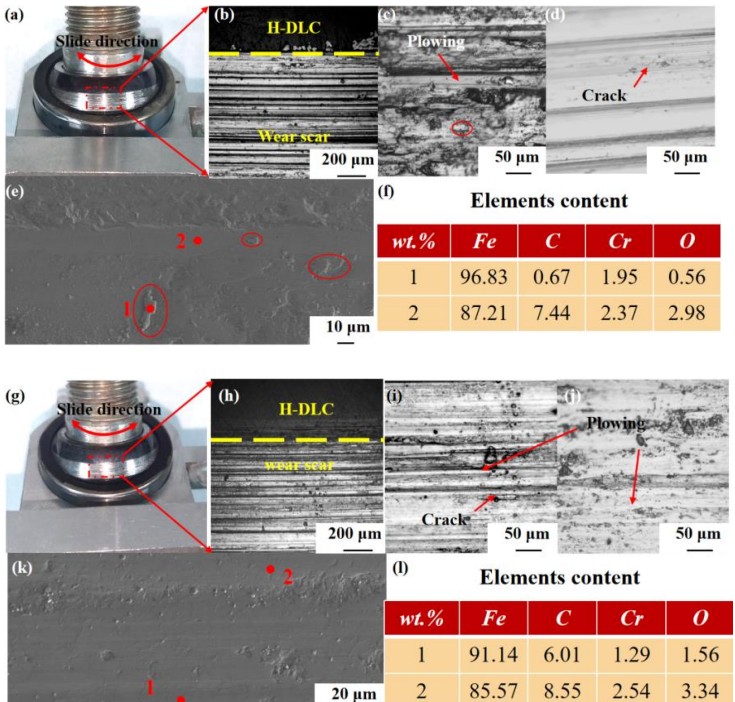

**Figure 10.** The optical image and EDS analyses of the wear zones of the H-DLC SSPBs in a vacuum environment: (**a**) wear zones of sample A; (**b**) optical image of sample A; (**c,d**) high magnification of sample A; (**e**) SEM morphology of sample A; (**f**) EDS analysis of sample A; (**g**) wear zones of sample B; (**h**) optical image of sample B; (**i,j**) high magnification of sample B; (**k**) SEM morphology of sample B; (**l**) EDS analysis of sample B.

## 4. Conclusions

In this paper, a tribometer was designed for testing coated SSPBs. The servo motor direct drive method with a simple structure and stepless speed regulation was selected. Considering the generation mechanism of the friction torque, a modular fixture scheme was used to realize the fixing of the inner and outer rings. Temperature and torque sensors were used to monitor the test bearing. Due to the vacuum environment, liquid cooling was adopted to control the test temperature. The influence of friction heat on self-made H-DLC SSPBs was studied in a vacuum environment. The variation trend of friction torque increased with a prolonged test time. In addition, the value of friction torque slightly fluctuated in the early stage but greatly fluctuated in the later stage because of wear. Based on the experimental data, the frictional heat had little effect on the H-DLC SSPBs' lifespans. In addition, a large number of groove morphologies and a small number of cracks indicated that the failure mechanisms of the H-DLC SSPBs were abrasive wear and fatigue wear. The development of this type of tribometer can be used for the mechanism analysis and performance evaluation of coated SSPBs.

**Author Contributions:** Conceptualization, Q.Y. and Z.L.; methodology, Z.Z.; software, A.W.; validation, G.M.; formal analysis, Z.Z.; inves-tigation, G.M.; resources, Z.Z.; data curation, Z.L.; writing—original draft preparation, Z.L.; writing—review and editing, Z.L.; visualization, H.W.; supervision, A.W.; project administration, H.W.; funding acquisition, G.M., Z.Z. and Q.Y. All authors have read and agreed to the published version of the manuscript.

**Funding:** This research was funded by the National Natural Science Foundation of China (Grant No. 52122508, 51905533), 145 Project, and State Key Laboratory of Mechanical System and Vibration Project (No. MSVZD202108).

**Data Availability Statement:** Not applicable.

**Conflicts of Interest:** The authors declare no conflict of interest.

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
