# Peer review of "Novel Tribometer for Coated Self-Lubricating Spherical Plain Bearings in a Vacuum"

_lubricants, doi:10.3390/lubricants10110291_

Round 1
Reviewer 1 Report
The authors have successfully developed a novel tribometer for investigating the tribological properties of coated self-lubricating spherical plain bearings in vaccum, which is innovative to some extent, but there are also the following problems:
1. In section introduction(Line 65 of the article),author sumed “through rationally selecting the dimension tolerance of the bearing substrate and the DLC films with excellent tribological properties”, However, there is no literature in this paper that clearance can improve the tribological performance.
2.The temperature sensor is installed on the outer ring fixture, which measures the outer ring temperature rather than the inner ring temperature, not the part with the highest bearing heat.
3.Is the temperature sensor contact measurement or non-contact measurement? How is it connected to the outer ring? Please give more details.
4.Is the tribometer suitable for other sizes of joint bearings? The size range of the test bearings should be listed in Table 1.
5. It is suggested to convert the friction torque into friction coefficient.
6. “Although the temperature had reached 48℃, the frictional heat had little effect on H-DLC SSPBs tribological properties. (line 21)Is this conclusion reasonable? Friction heat will cause the H-DLC coating to change, whether occurs graphitization, This will affect the tribological properties of bearings, and it is suggested that the authors do further research in this aspect.
7.It is found that the partially flat area(red circle) appeared(line 330). It is suggested that authors can make further analysis by EDS and other means.
8.During friction and wear, H-DLC coating exfoliate from the substrate would further participate in the wear process and accelerate the wear process of the materia, and these flakes aggravate the wear. The crack was found in sample B, but not in sample A. What is the main reason for the difference? it is suggested that the authors do further analysis in this aspect.
9.The conclusion should be more concise.
10.There were some spelling mistakes in the paper,such as “6. Sliding able (line 145),Radial force systme(line 180), In table2 Reciprocating requence” and so on , It is suggested that authors examine the full text carefully.
11.What are the inner ring base and outer ring material of the joint bearing? Please explain.
Reviewer 2 Report
· H-DLC full form is not given in the abstract.
· The structure of coated-SSPBs tribometer - needs more explanation.
Reviewer 3 Report
See attached file.

Round 2
Reviewer 3 Report
List of small questions/corrections:
1. When I wrote about “citations”, I meant, that you write “Citation: Li 1, Z.; Zhang 1, Z.; Yong 2, Q.; Ma 3, G.; Wei 4, A.; Wang 3,4 H. A Novel Tribometer for Coated Self-lubrication Spherical Plain Bearings in Vaccum. Lubricants 2022, volume number, x. https://doi.org/10.3390/xxxxx”, but it should not contain numbers (1,2,3..) or letters (a,b,c,..). This citation is between the lines 25 and 26, left side of the first page.
4. I steel do not understand, what means Ï• in the Table 1 in the part: “≤Ï•450mm×250mm”, you wrote the same in the answer: “Ï•450mm×250mm”.
9. What means “Real Time” (Test Time ≥ Real Time) in Figure 5?
Line 221: you can show the value 6E-6Pa as 6∙10-6 Pa (you are not working in the text file and no need to use E).
Please check Fig. 10 (you showed slide direction in the panel a-2, but this arrow should be shown in the panel a-1).
